# Ion channels and calcium signaling in motile cilia

**Julia F Doerner[1,2], Markus Delling[1,2], David E Clapham[1,2]\***

[1]Department of Cardiology, Howard Hughes Medical Institute, Boston Children's Hospital, Boston, United States; [2]Department of Neurobiology, Harvard Medical School, Boston, United States

**Abstract** The beating of motile cilia generates fluid flow over epithelia in brain ventricles, airways, and Fallopian tubes. Here, we patch clamp single motile cilia of mammalian ependymal cells and examine their potential function as a calcium signaling compartment. Resting motile cilia calcium concentration ([$Ca^{2+}$] ~170 nM) is only slightly elevated over cytoplasmic [$Ca^{2+}$] (~100 nM) at steady state. $Ca^{2+}$ changes that arise in the cytoplasm rapidly equilibrate in motile cilia. We measured $Ca_V1$ voltage-gated calcium channels in ependymal cells, but these channels are not specifically enriched in motile cilia. Membrane depolarization increases ciliary [$Ca^{2+}$], but only marginally alters cilia beating and cilia-driven fluid velocity within short (~1 min) time frames. We conclude that beating of ependymal motile cilia is not tightly regulated by voltage-gated calcium channels, unlike that of well-studied motile cilia and flagella in protists, such as *Paramecia* and *Chlamydomonas*.

**\*For correspondence:** dclapham@
enders.tch.harvard.edu

**Competing interest:** See
page 17

## Introduction

Cilia are ancient microtubule-based cellular appendages found in eukaryotic organisms (*Satir et al., 2008*). Motile cilia, like flagella, drive fluid flow via dynein-ATPase action on their 9 + 2 microtubular structure, while most primary cilia are solitary, nonmotile, and lack central microtubular pairs (9 + 0) (*Lindemann and Lesich, 2010*; *Satir and Christensen, 2007*). In mammals, almost all cells possess a single primary cilium that houses the Sonic Hedgehog pathway (*Drummond, 2012*) and mediates aspects of cell-cell signaling. Other nonmotile cilia are found in specialized sensory cells, such as rods and cones of the eye (*Satir and Christensen, 2007*). In contrast, multiple copies of motile cilia sprout from ependymal cells lining the brain ventricles, and epithelial cells in the airways and Fallopian tubes (*Brooks and Wallingford, 2014*; *Satir and Christensen, 2007*). A distinct type of motile cilium (9 + 0) additionally protrudes from cells in the embryonic node during development (*Babu and Roy, 2013*).

Motile cilia are much like eukaryotic flagella that drive locomotion in spermatozoa and protists (*Brooks and Wallingford, 2014*; *Satir and Christensen, 2007*). $Ca^{2+}$ influx modulates motility in the flagella of spermatozoa through specialized $Ca^{2+}$-selective, pH-sensitive CatSper channels to produce hyperactivated motility (*Kirichok et al., 2006*; *Miki and Clapham, 2013*; *Qi et al., 2007*; *Ren et al., 2001*). $Ca^{2+}$ influx also alters the flagellar waveform or ciliary beating direction in *Chlamydomonas* and *Paramecium*, respectively, and arrests beating in Mussel gill epithelia (*Bessen et al., 1980*; *Inaba, 2015*; *Naito and Kaneko, 1972*; *Tsuchiya, 1977*; *Walter and Satir, 1978*). Analysis of *Paramecium* and *Chlamydomonas* mutants and electrophysiological recordings identified voltage-gated calcium channels ($Ca_V$) in cilia/flagellar membranes as required regulators of ciliary beating (*Beck and Uhl, 1994*; *Dunlap, 1977*; *Fujiu et al., 2009*; *Kung and Naito, 1973*; *Matsuda et al., 1998*). These observations suggest a conserved $Ca^{2+}$ channel-dependent mechanism regulating flagellar/ciliary beating. Whether ion channels in motile cilia of mammalian cells changes their beat

**eLife digest** Certain specialized cells in the brain, airways and Fallopian tubes have large numbers of hair-like structures called motile cilia on their surface. By beating in a synchronized manner, these cilia help to move fluids across the surface of the cells: for example, cilia on lung cells beat to clear mucus away, while those in the brain help the cerebrospinal fluid to circulate.

Motile cilia in mammals are structurally similar to the flagella that propel sperm cells and certain single-celled organisms around their environments. These flagella have specialized pore-forming proteins called ion channels in their membrane through which calcium ions can move. This flow of calcium ions controls the beating of the flagella. However, it is unclear whether a similar movement of calcium ions across the cilia membrane regulates motile cilia beating in mammals.

Doerner et al. have now used a method called patch clamping to study the movement of calcium ions across the membrane of the motile cilia found on a particular type of mouse brain cell. This revealed that unlike flagella, these motile cilia have very few voltage-gated calcium channels; instead, the vast majority of these ion channels reside in the main body of the cell. Furthermore, the level of calcium ions in the motile cilia follows changes in calcium ion levels that originate in the cell body.

Overall, Doerner et al. demonstrate that the activity of voltage-gated calcium channels does not control the beating rhythm of the motile cilia in the mouse brain or how quickly the fluid above the cell surface moves. Future work should investigate whether this is also the case for the cells that line the trachea and Fallopian tubes.

frequency is not clear, but intraciliary $[Ca^{2+}]$-dependent changes in motile cilia beating has been reported by several groups (*Di Benedetto et al., 1991*; *Girard and Kennedy, 1986*; *Lansley et al., 1992*; *Nguyen et al., 2001*; *Schmid and Salathe, 2011*; *Verdugo, 1980*). The question that we seek to answer in this study is whether $Ca^{2+}$-permeant ion channels are present in motile cilia, and if so, do they change motile cilia behavior.

Successful whole-cilia patch clamping of fluorescently-labeled immotile primary cilia revealed non-selective cation currents (PKD2-L1 + PKD1-L1 heteromeric complexes) in primary cilia membranes (*DeCaen et al., 2013*; *Delling et al., 2013*). Here, we examined ion currents in fluorescently-labeled, voltage-clamped motile cilia of brain ependymal cells and demonstrate that motile cilia are well coupled electrically and by diffusion to the cellular compartment. We show that few $Ca_V$ channels are present in the cilia membrane, that resting $[Ca^{2+}]$ is only slightly elevated in motile cilia, and that motile cilia $[Ca^{2+}]$ is driven primarily by changes in cytoplasmic $[Ca^{2+}]$. Excitation of the ependymal cell by membrane depolarization increases ciliary $[Ca^{2+}]$ with only minor changes in motility and fluid movement, suggesting that beating of ependymal motile cilia is not significantly regulated by the activity of ciliary or cytoplasmic $Ca_V$ channels.

## Results

### Ependymal motile cilia identification and patch clamp

We initially examined ependymal cell GFP-labeled motile cilia from immunolabeled brain sections of transgenic *Arl13b-EGFP$^{tg}$* mice (*Delling et al., 2013*). Ciliary localization of Arl13b-EGFP was confirmed by co-staining with the ciliary marker, acetylated tubulin (*Figure 1A*). We also observed GFP-labeled motile cilia in primary cultures. At day 10 in vitro (DIV10), ~88% of acetylated tubulin stained multiciliated ependymal cells (n = 400 cells) had GFP-labeled cilia (n = 352 cells). Transgene expression varied, with ~one-third of the cells exhibiting weak GFP fluorescence in cilia (*Figure 1B*, arrow). Some cells had only a few motile cilia (either mono-, bi- or sparsely ciliated) that often displayed moderate motility (~two–three fold slower beat frequency) as compared to neighboring multiciliated cells (*Figure 1B*, arrow head). Sparsely ciliated cells may be the result of ependymal cell maturation, in vitro culturing, or represent a population of previously reported biciliated ependymal cells (*Mirzadeh et al., 2008*). Sparse cilia likely experience different hydrodynamic forces than large

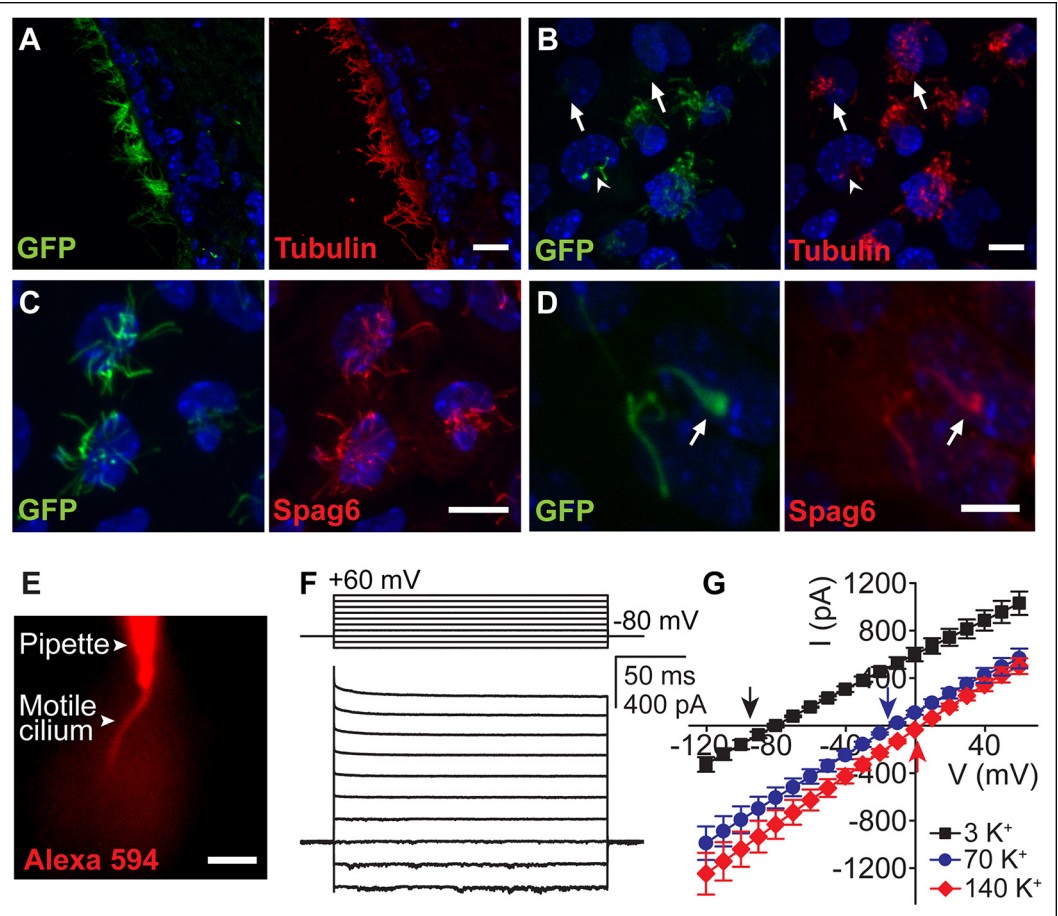

**Figure 1.** Ependymal motile cilia identification and patch clamp. (**A**) Immunolabeling of motile cilia in a section of the lateral ventricle from an *Arl13B-EGFP^tg* mouse. Ciliary localization of Arl13B-EGFP was confirmed using anti-GFP (green) and anti-acetylated tubulin antibodies (red). (**B**) Anti-GFP (green) and anti-acetylated tubulin staining (red) of cultured ependymal cells at DIV10. GFP labeling of motile cilia varied, with some cells displaying only weak or barely detectable GFP fluorescence in cilia (arrows). Some cells were only sparsely ciliated (arrowhead). (**C**) Staining of cultured multiciliated cells with anti-GFP (green) and anti-Spag6 (red). (**D**) Representative staining of a previously recorded ependymal cell grown on a gridded glass bottom dish (grid size, 50 μm, arrow marks motile cilium). Motile cilia of sparsely ciliated cells were GFP (green) and Spag6 (red) positive (n = 40/45). Nuclei were labeled with Hoechst dye (blue, **A–D**). Panels **B-D** display average intensity z-projections of image stacks. Scale bars, 10 μm (**A–C**) and 5 μm (**D**). (**E**) Image showing dye diffusion into a motile cilium after successful break-in (50 μM Alexa 594 hydrazide, n = 9). Scale bar, 3 μm. (**F**) Example current (bottom) recorded in the whole-motile-cilium configuration in response to increasing voltage steps (top). Holding potential, -80 mV. (**G**) Mean steady state current after break-in plotted as a function of command voltage (n = 4). External solution (aCSF) with 3 mM KCl (black filled squares), 70 mM KCl (blue filled circles), and 140 mM KCl (red filled diamonds). Arrows in the graph indicate calculated $E_K$ values. Error bars; ± SEM.

groups of synchronized cilia in multiciliated cells (metachronism), which could explain differences in motility (*Guirao and Joanny, 2007*; *Guirao et al., 2010*).

Motility and multiciliation hinder patch clamp recordings from individual cilia and thus recordings were from sparsely ciliated cells. We verified the 9 + 2 structural arrangement of the cilia by imaging the cells on coded, gridded glass dishes and subsequent staining (*Video 1*). Spag6, a protein associated with the central pair of microtubules (*Sapiro et al., 2002*), was detected in motile cilia of multi- and sparsely-ciliated cells (*Figure 1C,D*). Eighty-nine percent of recorded sparsely ciliated cells exhibited Spag6 immunoreactivity in their motile cilia (*Figure 1D*; n = 40/45 cells), consistent with the presence of a central pair of microtubules. In contrast, only a small number of primary cilia from

mIMCD cells were (weakly) immunofluorescent in control experiments (8%, n = 9/114, data not shown).

Typically, we patched the bulging tip of a motile cilium. High resistance seals were obtained by positioning the patch electrode in the focal plane at a position near the end of the cilia's stroke, applying suction at the moment the cilium approached the pipette tip (*Video 2*). Access to a motile cilium was typically achieved by applying a series of brief voltage pulses. Upon successful 'break-in', we were able to monitor dye diffusion from the electrode into the cilium (*Figure 1E*). Recording via the whole-motile-cilium yielded a linear current that reversed at hyperpolarized potentials (*Figure 1F,G*). These currents were primarily carried by $K^+$ (*Figure 1G*), reminiscent of the ohmic currents reported for ependymal cell bodies (*Genzen et al., 2009a*; *Liu et al., 2006*; *Nguyen et al., 2001*).

## Electrical coupling of motile cilia to the cellular compartment

To determine whether ependymal motile cilia are electrically coupled to the cellular compartment, we measured their electrical properties and compared them to whole-cell recordings (*Figure 2A,B*). Input resistances immediately upon break-in were on average ~five-fold higher in whole-cilium recordings (~180 MΩ, n = 8) than in whole-cell recordings (~36 MΩ, from the cell body and connected cells, n = 12), reflecting the resistance introduced by the cilium (see below). Uncoupling the cells with flufenamic acid (FFA), an anthranilic acid derivative known to inhibit gap junctions (*Harks et al., 2001*), decreased the measured conductance in both whole-cell and whole-cilium recordings. Bath substitution with TEA-Cl/BaCl$_2$ blocked the remaining $K^+$ conductances (*Figure 2A, B*).

To directly assess coupling between motile cilia and cell compartments, we analyzed the capacitive current recorded under voltage clamp in the whole-cell and whole-motile-cilium configuration. In response to a hyperpolarizing voltage step, capacitive current relaxation was 8–10 times faster in whole-cell recordings (0.3 ± 0.1 ms in aCSF/FFA and 0.4 ± 0.1 ms in TEA-Cl/BaCl$_2$) than in whole-motile-cilium recordings (3.1 ± 0.3 ms in aCSF/FFA and 3.3 ± 0.4 ms in TEA-Cl/BaCl$_2$; *Figure 2C,D*). By integrating the current transient to yield net capacitance, we determined that a similar surface area was charged in whole-cell (20.4 ± 3.2 pF in aCSF/FFA, 18.5 ± 2.5 pF in TEA-Cl/BaCl$_2$) and whole-motile-cilium recordings (18.3 ± 2.4 pF in aCSF/FFA, 19.0 ± 2.8 pF in TEA-Cl/BaCl$_2$; *Figure 2E*). Calculating the series resistance from the time constant and membrane capacitance, we determined a nine–ten fold higher resistance in whole-motile-cilia recordings (180.3 ± 20.6 MΩ in aCSF/FFA and 188.2 ± 25.0 MΩ in TEA-Cl/BaCl$_2$) as compared to whole-cell recordings (18.0 ± 3.4 MΩ in aCSF/FFA and 20.2 ± 3.7 MΩ in TEA-Cl/BaCl$_2$; *Figure 2F*). Together these findings suggest that motile cilia are indeed electrically coupled to the cellular compartment

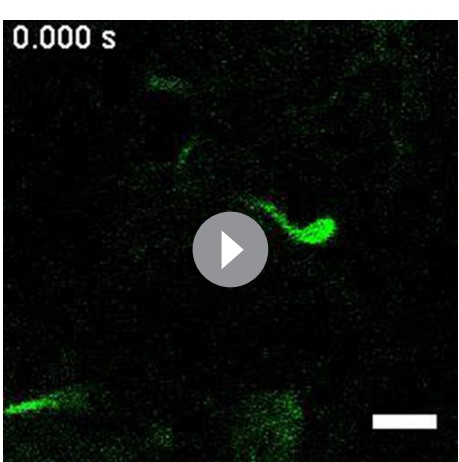

**Video 1.** Time lapse of a ciliated cell. Ependymal cells were grown on gridded culture dishes (grid size, 50 μm). Sparsely ciliated cells were recorded, fixed and stained with a central pair marker (Spag6). The example movie corresponds to the staining shown in *Figure 1D* (same grid). Frame rate 0.065 s, playback 1x. Scale bar, 5 μm.

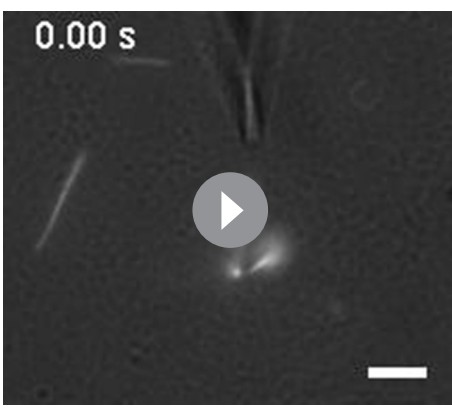

**Video 2.** Patch clamping of a motile cilium. Frame rate 0.58 s, playback 1x. Scale bar, 3 μm.

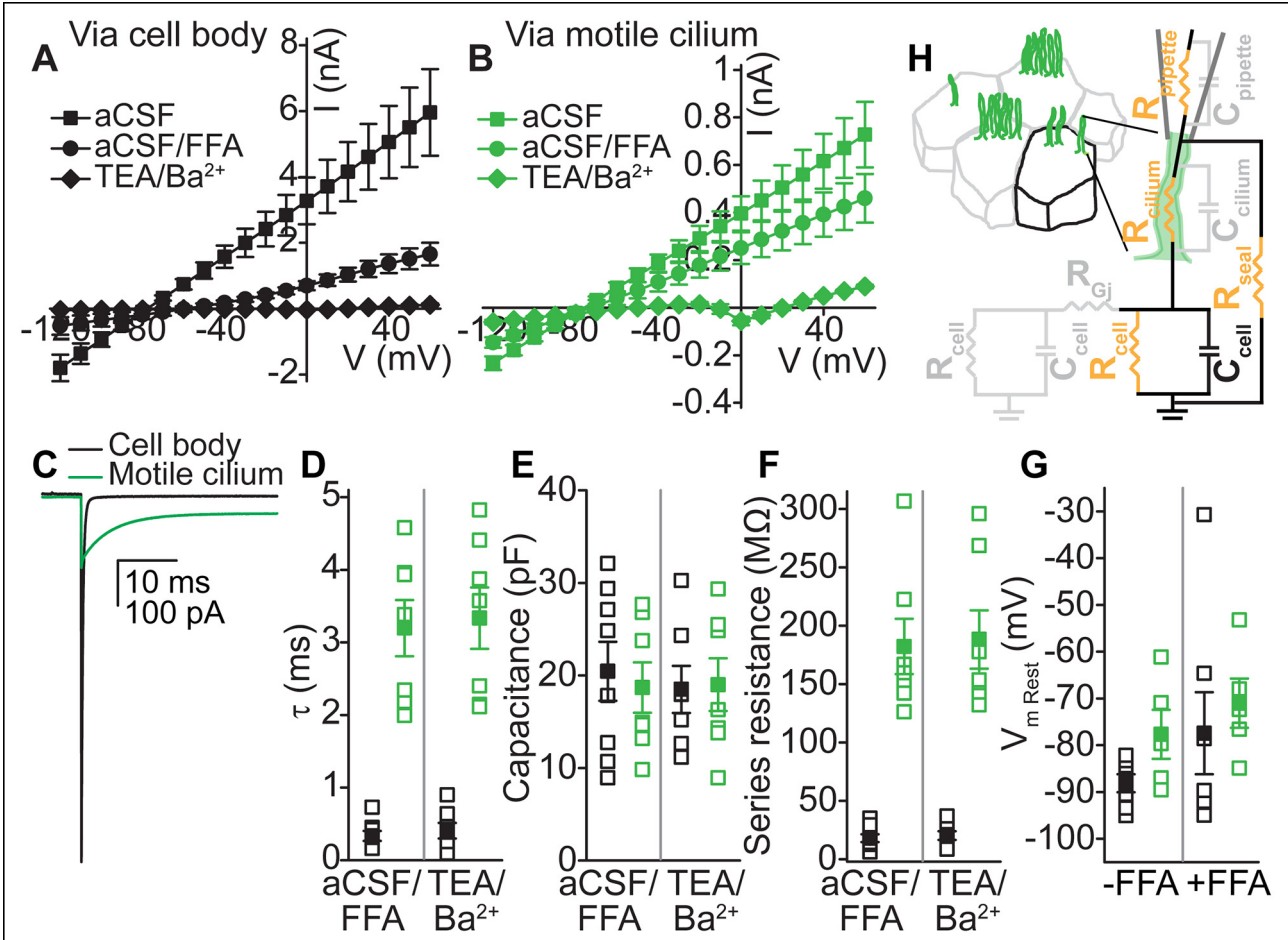

**Figure 2.** Electrical coupling of motile cilia to the cellular compartment. (**A,B**) Mean current-voltage relation recorded from the cell body (**A**, n = 6) or a motile cilium (**B**, n = 5) after break-in in aCSF (filled squares), subsequent cell uncoupling with flufenamic acid (FFA, 100 µM, 2 min, filled circles), or block of $K^+$ conductances in TEA-Cl/BaCl$_2$ (TEA/Ba$^{2+}$, filled diamonds; only cell/cilium recordings with input resistance >1 GΩ after TEA-Cl/BaCl$_2$ treatment are plotted). Holding potential between 200 ms steps, -80 mV. Note: The voltage in **A** and **B** refers to the command voltage. The voltage error, that is, the difference between the command voltage and membrane voltage produces a large error due to the high resistance through the cilium in series with the low, multicellular membrane resistance (resting $K^+$ conductance, cell-cell connections via gap junctions). Thus, large currents are inaccurate: the top traces only serve to show that flufenamic acid uncouples cells. (**C**) Example capacitive currents recorded in response to a 20 mV hyperpolarizing voltage step (50 ms) for the cell body (black) and motile cilium (green) after uncoupling with flufenamic acid (FFA, 100 µM) and block of $K^+$ conductances (TEA-Cl/BaCl$_2$). The steady state (time-independent) current in the motile cilium trace is leak current. (**D–F**) Time constant (**D**), membrane capacitance (**E**), and series resistance (**F**) determined from an average of 100–200 sweeps of capacitive current for cell body (black squares) and motile cilium (green squares) recordings after cell uncoupling with flufenamic acid (FFA, 100 µM, ~5 min) and perfusion with TEA-Cl/BaCl$_2$ (TEA/Ba$^2$$^+$, n = 7–8). (**G**) Resting membrane potential assessed under current clamp with a gramicidin-perforated patch for cell bodies (black squares) and motile cilia (green squares) before (n = 7 cell body, n = 8 motile cilia) and after (n = 7 cell body, n = 5 motile cilia) addition of flufenamic acid (FFA, 100 µM) to the bath. Open squares represent the range of individual cells/cilia; filled squares are the mean. Error bars; ± SEM. (**H**) Cartoon illustrating simplified equivalent circuit of access to the cellular compartment via a motile cilium. The cable-like properties of a motile cilium significantly increase the access (series) resistance. Block of gap junctions by flufenamic acid removes the contributions from neighboring cells.

via a high resistance cable determined by the length and cross-sectional area of a motile cilium (estimated as ~350 MΩ for an ideally insulated cable; *Figure 2H*, see Materials and methods). In other words, currents recorded in the whole-motile-cilium configuration can be attributed to channel openings in the cell and/or cilia membrane. Finally, we measured an only slightly depolarized resting membrane potential for motile cilia as compared to the ependymal cell body (-77 ± 3 mV, motile cilia; -88 ± 2 mV, cell body; *Figure 2G*).

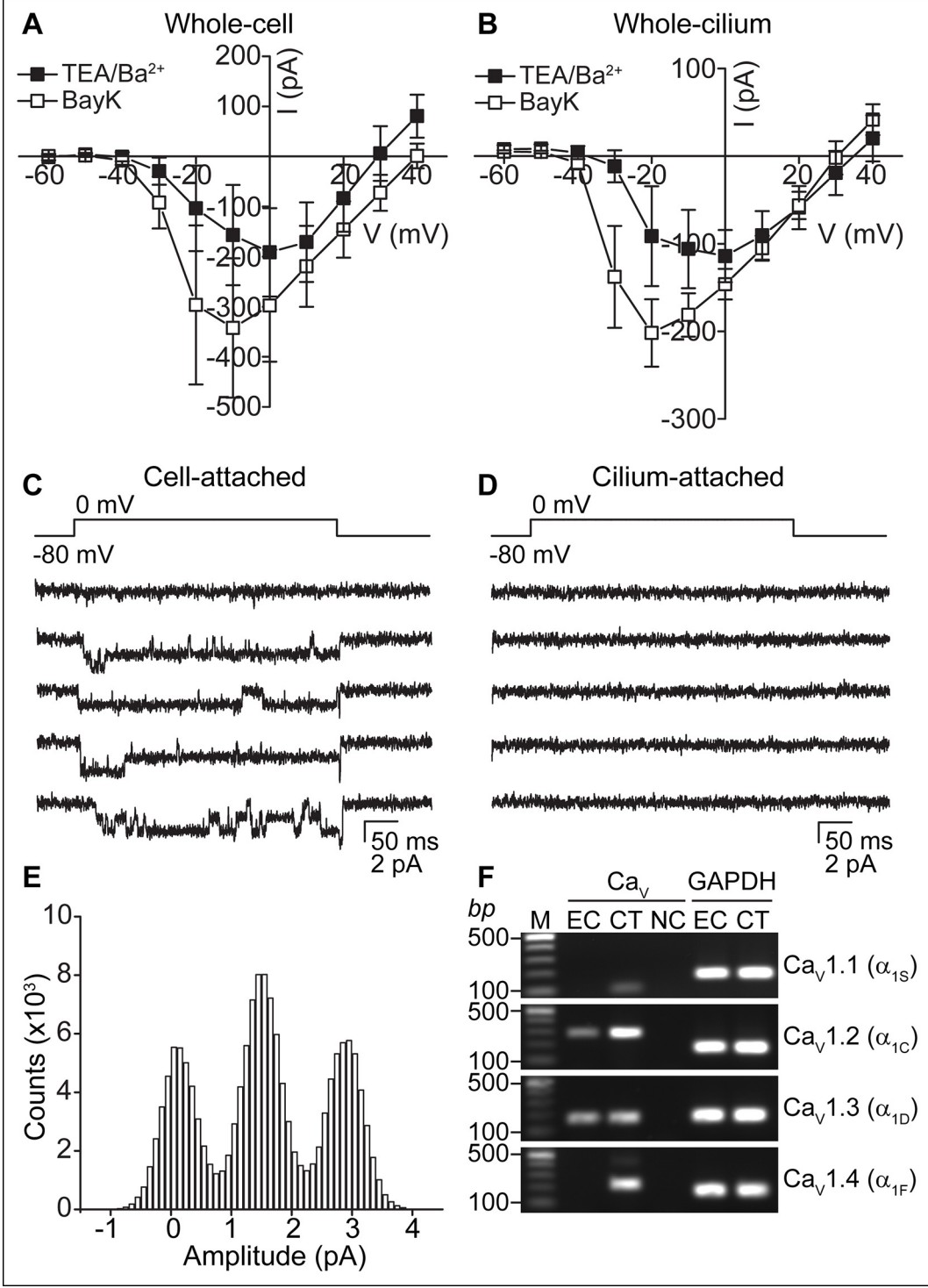

**Figure 3.** $Ca_V$-mediated currents and single channels in ependymal cells. (**A,B**) Average peak current in response to voltage steps (500 ms) before (filled squares, in TEA-Cl/BaCl$_2$) and after addition of the $Ca_V$ potentiator, BayK 8644 (BayK, 5 µM, open squares), recorded from the cell body (**A**, n = 7) or motile cilia (**B**, n = 8). Peak current amplitudes varied substantially (range pre BayK treatment: cells, -58 pA to -726 pA; cilia, -25 pA to -413 pA). The high series resistance in whole-motile-cilium recordings shifts the peak to more hyperpolarized potentials. Holding potential, -80 mV. Cells were uncoupled by flufenamic acid (FFA, 100 µM). Error bars; ± SEM. (**C,D**) Example of 5 consecutive traces recorded in cell-attached (**C**) or cilium-attached (**D**, pipette filled with BaCl$_2$). BayK-induced long lasting $Ca_V$ channel openings were observed in 6 of 8 cell-attached recordings (BayK, 5 µM, bath). $Ca_V$

*Figure 3. continued on next page*

*Figure 3. Continued*

channel openings were rare in motile cilia-attached recordings (n = 1/29, see *Figure 3—figure supplement 1*; note: smaller pipettes in motile cilia recordings results in smaller membrane area sampling). (**E**) All point amplitude histogram of all traces from the recording shown in **C**. (**F**) RT-PCR showing amplification of $Ca_V1.2$ and $Ca_V1.3$ transcripts from cDNA derived from cultured ependymal cells (EC, DIV10). cDNAs from skeletal muscle ($Ca_V1.1$), heart ($Ca_V1.2$), brain ($Ca_V1.3$), and eyes ($Ca_V1.4$) served as positive control tissues (CT). Minus reverse transcriptase negative control (NC). Molecular ladder (M). GAPDH was amplified from all cDNAs. Images cropped for illustration.

The following figure supplements are available for Figure 3:

**Figure supplement 1.** $Ca_V$-mediated currents and single channels in ependymal cells.

## $Ca_V$-mediated currents in ependymal cells

When $K^+$ currents were blocked in the $TEA-Cl/BaCl_2$ bath, we frequently observed relatively small voltage-dependent inward currents (*Figure 2B*). Consistent with the finding that cilia and cellular compartments are electrically coupled, we recorded $Ca_V$ currents in both whole-cell and whole-cilium configurations (*Figure 3A,B*; high series resistance in whole-motile-cilium recordings affected the time course and peak of the calcium current; see Materials and methods). These currents were potentiated by BayK8644 and reduced by nimodipine and $CdCl_2$ (*Figure 3A,B* and *Figure 3—figure supplement 1A*), consistent with the pharmacology of L-type calcium channels ($Ca_V1$ subfamily) (*Catterall et al., 2005*). Indeed, $Ca_V1.2$ and $Ca_V1.3$ α subunit transcripts were detected in the cDNA of cultured ependymal cells (DIV10; *Figure 3F*).

To address the question of whether $Ca_V$ currents are in motile cilia membranes, cell body membranes, or both, it would be ideal to detach the cilium from the cell. However, we were unable to detach and record from isolated motile cilia, as we were able to do in primary cilia (*DeCaen et al., 2013*). Thus, we recorded from cell and motile cilia membranes in the membrane-attached configuration. Prolonged single channel openings in the presence of BayK8644 averaged 1.4 pA at 0 mV (from the record shown in *Figure 3C*) and were frequently observed in cell-attached recordings, but were typically not detected in cilium-attached recordings (*Figure 3C–E*). Despite many attempts, brief single channel openings reminiscent of L-type channel openings in the absence of BayK8644 (*Hess et al., 1984*) were recorded from only a single motile cilium patch (*Figure 3—figure supplement 1C,D*). Full-length ciliary recordings (*Kleene and Kleene, 2012*), were also not feasible given the presence of $Ca_V$ channels in the cell membrane. Nevertheless, the low percentage of motile cilium patches in which we observed $Ca_V$ channel openings suggests that $Ca_V$ channels are not enriched in ependymal motile cilia. If channel densities are identical in cell and cilia membranes, we would only expect ~3 channels in a motile cilium (see Materials and methods). These $Ca^{2+}$-permeant ion channels and currents in motile cilia are clearly distinct from the nonselective currents recorded from primary cilia (*DeCaen et al., 2013*).

## Motile cilia [$Ca^{2+}$] can be modified by cytoplasmic [$Ca^{2+}$]

To further examine the potential functional consequences of $Ca_V$ channel activity, we evaluated motile cilia [$Ca^{2+}$] by targeting a genetically encoded ratiometric $Ca^{2+}$ sensor to these cilia (*Delling et al., 2013*). Previous work established Somatostatin Receptor 3 (SSTR3) transgene expression in ependymal motile cilia (*O'Connor et al., 2013*). We fused GCaMP6s (*Chen et al., 2013*) with mCherry to the C-terminus of SSTR3 and transduced cultured ependymal cells with a recombinant adenoviral vector (pAd-mSSTR3-mCherry-GCaMP6s), resulting in abundant expression of the sensor in motile cilia (*Figure 4A*). Addition of ionomycin evoked a rapid increase in GCaMP6s fluorescence, confirming the ability of the sensor to report changes in ciliary [$Ca^{2+}$] (*Figure 4B,C*). We calibrated the ratiometric sensor (see Materials and methods) and determined the average ratio of F_GCaMP6s and F_mCherry in motile cilia under basal conditions (aCSF, 1.4 mM extracellular $Ca^{2+}$; *Figure 4D, E*). To avoid offsets in the position of beating cilia, we scanned non-sequentially and corrected ratios for bleed-through (*Figure 4—figure supplement 1A,B*, and Materials and methods). Consistent with a relatively low number of $Ca_V$ channels and a hyperpolarized resting membrane potential (at which $Ca_V$ channels are inactive), we determined the resting motile cilia [$Ca^{2+}$] as 165 nM (average

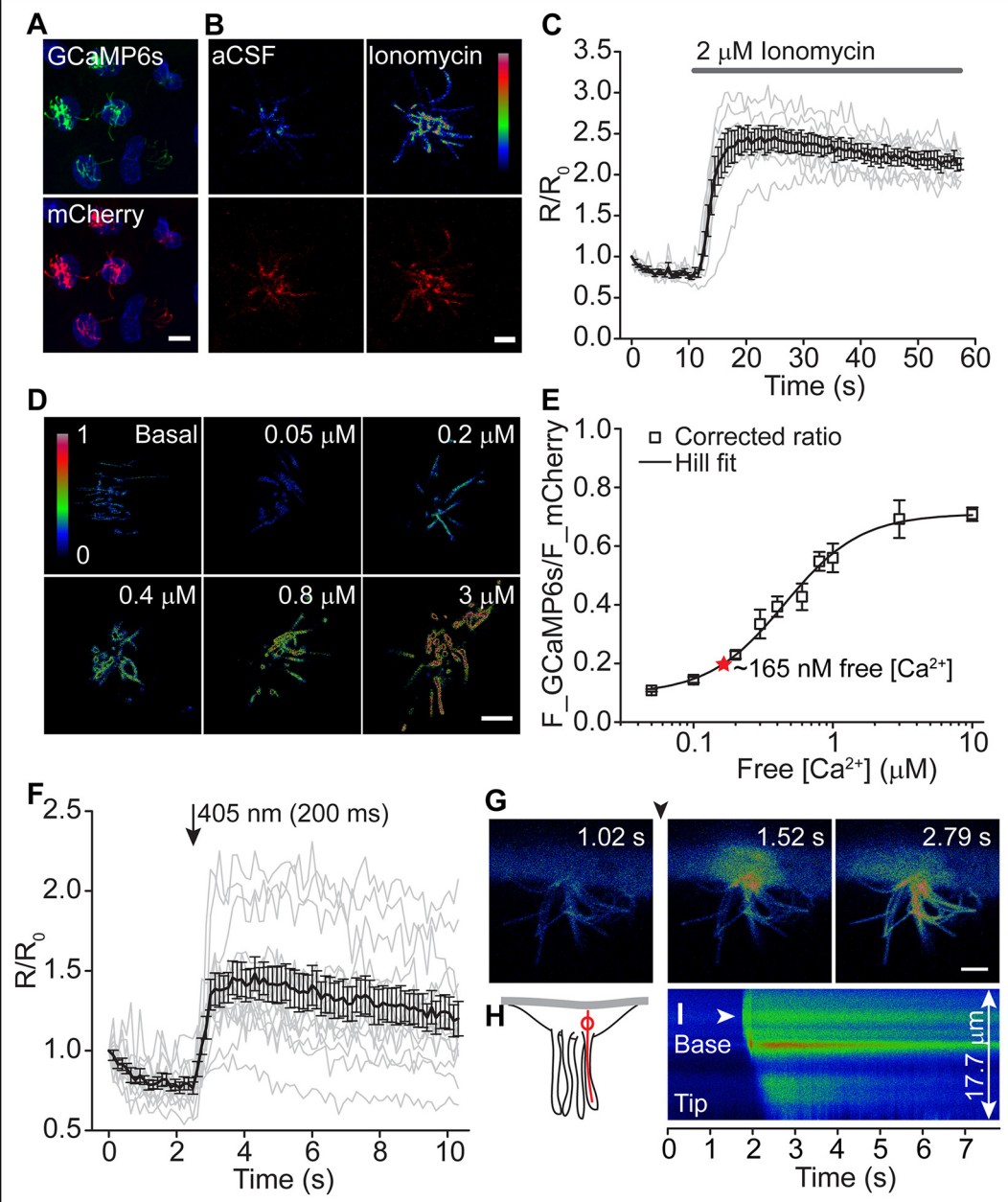

**Figure 4.** Motile cilia [Ca²⁺] can be modified by cytoplasmic [Ca²⁺]. (**A**) Cluster of recombinant adenovirus-transduced ependymal cells expressing the cilia-targeted fusion construct mSSTR3-mCherry-GCaMP6s. In fixed cells, cilia were recognized by staining with anti-GFP (green) and anti-mCherry (red) antibodies. (**B,C**) mSSTR3-mCherry-GCaMP6s reported changes in motile cilia [Ca²⁺] in response to ionomycin (2 µM). Example images (**B**) showing GCaMP6s (pseudocolor) and mCherry fluorescence before (in aCSF) and after addition of ionomycin to the bath, and quantified ratio changes (**C**, n = 7 cells). The ratio of GCaMP6s and mCherry (R) was normalized to the initial ratio (R₀). (**D**) Example pseudocolor images of F_GCAMP6s/F_mCherry ratios of ependymal motile cilia in aCSF (basal) and under defined free [Ca²⁺] in the bath. Ratio images were background-subtracted and thresholded. (**E**) Calibration curve showing F_GCAMP6s/F_mCherry ratio plotted as a function of free [Ca²⁺] (n = 5–8 for each [Ca²⁺]). Resting motile cilia [Ca²⁺] was 165 nM at steady state (n = 30, red star). (**F**) Quantified changes in F_GCAMP6s/F_mCherry ratio in response to Ca²⁺ uncaging in the cytoplasm (n = 13). (**G**) Example pseudocolor images from a time lapse recording of an ependymal cell, recorded from the side (see Materials and methods). Ca²⁺ was uncaged in the cytoplasm at the cilia base (approx. time point marked by arrowhead, 405 nm illumination for 200 ms). (**H**) Cartoon illustrating line scanning. The red circle and red line indicate the typical position of the uncaging stimulus and line scan. (**I**) Example record of a line scan through the

*Figure 4. continued on next page*

*Figure 4. Continued*

cytoplasm and a motile cilium displayed in pseudocolor. The arrowhead marks the position of uncaging at the ciliary base. $Ca^{2+}$ rapidly diffused from the cytoplasm into the motile cilium ($593 \pm 86$ ms to the tip, n = 15). Error bars; $\pm$ SEM. Scale bars, 10 µm (**A**) and 5 µm (**B,D,G**).

The following figure supplements are available for Figure 4:

**Figure supplement 1.** Motile cilia [$Ca^{2+}$] can  be modified by cytoplasmic [$Ca^{2+}$].

ratio: $0.2 \pm 0.01$; *Figure 4D, E*). In control experiments in which immobilized cilia were scanned sequentially, a similar resting [$Ca^{2+}$] was determined (167 nM, *Figure 4—figure supplement 1C*, and see below). We conclude that motile cilia resting [$Ca^{2+}$] is only slightly elevated in comparison to cytoplasmic [$Ca^{2+}$] (~100 nM) (*Clapham, 2007*) at steady state, in contrast to the ~500 nM elevation observed in primary cilia (*Delling et al., 2013*).

To assess $Ca^{2+}$ diffusion between the cell body and motile cilia, we loaded transduced ependymal cells with caged $Ca^{2+}$ (NP-EGTA-AM). A brief uncaging stimulus in the cytoplasm rapidly increased ciliary GCaMP6s fluorescence (*Figure 4F*). However, imaging motile cilia through the z-axis of the cilia (that is, top or bottom views) prevented further analysis of $Ca^{2+}$ diffusion. In order to track $Ca^{2+}$ waves, we halted ciliary beating and imaged ependymal cilia from the side (*Figure 4G,H*, see Materials and methods). As reported previously, sodium metavanadate (SMVD) treatment greatly reduced motility, presumably through inhibition of the dynein ATPase (*Gibbons et al., 1978*; *Nakamura and Sato, 1993*) (*Figure 4—figure supplement 1D,E*). Upon uncaging at the ciliary base (visualized by additional loading of the cells with the $Ca^{2+}$ indicator Oregon Green 488 BAPTA-1 AM, OGB-1), $Ca^{2+}$ entered the cilium and moved from the base to the tip with an apparent velocity of $22 \pm 3$ µm/s (*Figure 4G–I*, *Video 3*). Thus, fluctuations in cytoplasmic [$Ca^{2+}$] can readily diffuse across the cell-cilia boundary to modify motile cilia [$Ca^{2+}$], as we found for primary cilia (*Delling et al., 2013*).

## Depolarization increases ciliary [$Ca^{2+}$], but not ciliary beat frequency or fluid velocity

We next asked whether activation of $Ca_V$ channels increases ciliary [$Ca^{2+}$] and regulates ciliary motility in ependymal cells. Differences in the rise times of the two $Ca^{2+}$ indicators (OGB-1 and GCaMP6s) precluded a meaningful analysis of signal onsets in motile cilia versus the cytoplasm. The homogeneity of voltage in cell body and cilium, as well as the observed rapid diffusion of $Ca^{2+}$ from the cytoplasm into motile cilia, suggests that ciliary [$Ca^{2+}$] will increase quickly, irrespective of the localization of $Ca_V$ channels (*Figure 4*). By recording ependymal cells in current clamp, we measured the response to increasing external [$K^+$] and observed a depolarization of membrane potential from $-87 \pm 3$ mV at rest (aCSF 3 $K^+$) to $-48 \pm 5$ mV, $-31 \pm 1$ mV, $-18 \pm 0.7$ mV, and $-1.7 \pm 2$ mV in aCSF 20 $K^+$, 40 $K^+$, 70 $K^+$, or 140 $K^+$, respectively (*Figure 5—figure supplement 1A, B*). Bath perfusion of depolarizing external [$K^+$] (>40 mM [$K^+$]) robustly increased cytoplasmic and ciliary [$Ca^{2+}$] (*Figure 5A,B* and *Figure 5— figure supplement 1C,D,F*). The response was greatly reduced by pre-incubation with nimodipine and was absent in $Ca^{2+}$-chelated bath saline (calculated as ~2.5 nM free [$Ca^{2+}$]

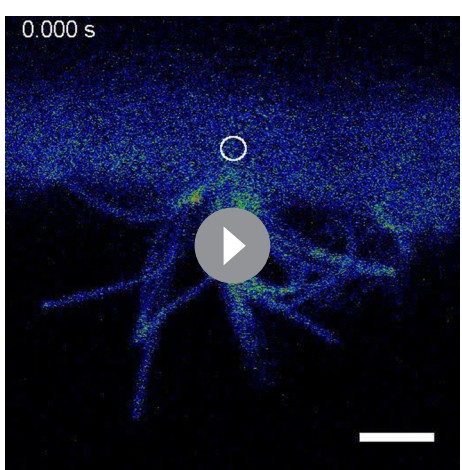

0.000 s

**Video 3.** Time lapse of a cell imaged from the side. $Ca^{2+}$ was uncaged in the cytoplasm at the cilia base (approximate location indicated by circle) with a brief 405 nm laser pulse (time point 1.016 s for 200 ms) and diffused into the motile cilia. Cilia movement was inhibited by sodium metavanadate (100 µM). Frame rate 0.254 s, playback 1x. Scale bar, 5 µm.

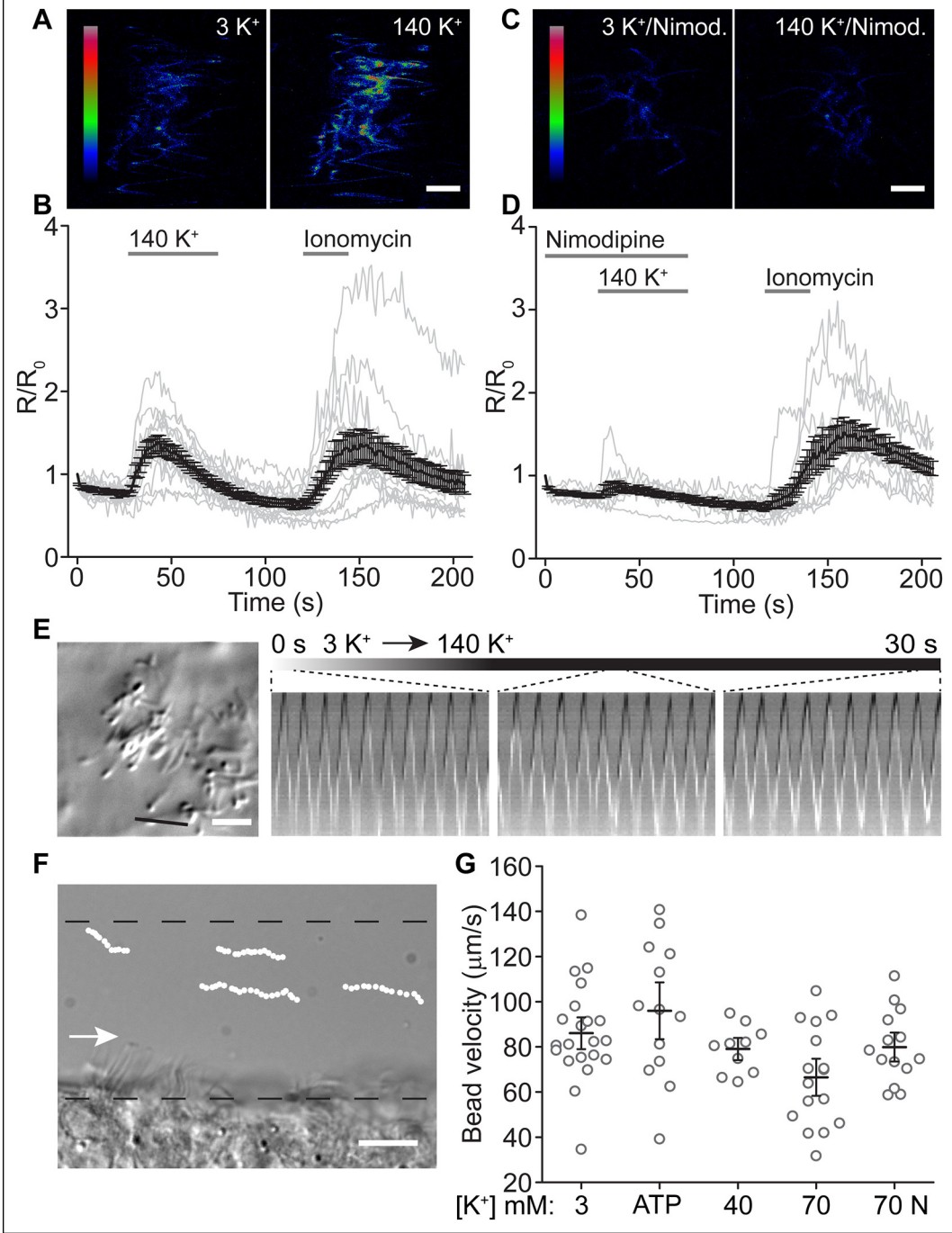

**Figure 5.** Depolarization increases ciliary [Ca²⁺], but not ciliary beat frequency or fluid velocity. (A,B) Example pseudocolor images of a cell expressing the ratiometric sensor mSSTR3-mCherry-GCaMP6s in motile cilia before and during perfusion of 140 K⁺ (A) and quantification of ciliary F_GCaMP6s/ F_mCherry ratio changes in response to 140 K⁺ (B, n = 15 cells). Ionomycin (1 μM) was applied as control stimulus. (C,D) Example pseudocolor images of a cell pre-treated with nimodipine (10 μM, 2 min, C) and quantification of ratio changes in response to a depolarizing stimulus (140 K⁺) after Ca$_V$ channel block by 10 μM nimodipine (D, n = 11 cells). (E) Ciliary beat frequency of cultured ependymal cells was not substantially altered during perfusion of depolarizing [K⁺] solutions (70 or 140 K⁺, n = 5 coverslips). The black line in the DIC image (left) indicates the position of the line used to derive the kymographs. Kymographs were analyzed at the indicated time points (duration, 1 s). (F) Example image of a brain slice showing frame by frame position of tracked beads along the lateral ventricle. Dotted lines indicate the area in which beads were tracked (<30 μm from surface). The arrow marks the direction of fluid flow. (G) Bead velocities measured under the conditions indicated. The mean is indicated by the black line and the open circles (gray) represent the velocities of all tracked beads for each condition (3 K⁺: n = 6 slices, 21 beads; ATP (100 μM): n = 4 slices, 13 beads; 40 K⁺: n = 2 slices, 10 beads; 70 K⁺: n = 5 slices, 16 beads; 70 K⁺ + 10 μM nimodipine (70 N): n = 5 slices, 14 beads). Error bars; ± SEM. Scale bars, 5 μm (A,C,E) and 10 μm (F).

*Figure 5. continued on next page*

*Figure 5. Continued*

The following figure supplements are available for Figure 5:

**Figure supplement 1.** Depolarization increases ciliary [$Ca^{2+}$], but not ciliary beat frequency or fluid velocity.

assuming ~10 µM total $Ca^{2+}$; *Figure 5C,D* and *Figure 5—figure supplement 1E*), suggesting that $Ca_V$ channel activation accounts for the increase in ciliary [$Ca^{2+}$].

Previous studies suggest that $Ca^{2+}$ increases the beating frequency of motile cilia in airways and Fallopian tubes (*Di Benedetto et al., 1991*; *Girard and Kennedy, 1986*; *Lansley et al., 1992*; *Schmid and Salathe, 2011*; *Verdugo, 1980*). Nguyen et al, reported a serotonin-mediated $Ca^{2+}$-dependent increase in ciliary beating frequency of ependymal motile cilia (*Nguyen et al., 2001*). We thus examined ciliary beating in response to depolarizing external [$K^+$] by plotting kymographs of individual cilia (*Lechtreck et al., 2008*). We observed no significant difference in the beating frequency during perfusion of depolarizing external [$K^+$] solution (aCSF 70 $K^+$ or aCSF 140 $K^+$). For example, cilia beat frequency was initially 9.7 ± 1 Hz (0–1 s) and was unchanged by perfusion with depolarizing high [$K^+$] solution (9.7 ± 1 Hz at 15–16 s and 9.3 ± 0.8 Hz at 29–30 s; *Figure 5E*, see Materials and methods). For comparison cilia beat frequency during continuous perfusion of standard external solution (aCSF 3 $K^+$) was 9.8 ± 1 Hz (0–1 s), 10 ± 1 Hz (15–16 s), and 10 ± 1 Hz (29–30 s). Kymographs depicting the beat cycles of individual cilia additionally revealed that cilia smoothly transitioned from a forward to a backward stroke independent of external [$K^+$] (*Figure 5E*).

While ciliary beating was unaffected in vitro, we next asked whether activation of $Ca_V$ channels by a depolarizing stimulus would disturb fluid movement in acute slices of the lateral ventricles. We assessed the velocity of polystyrene beads (500 nm) in proximity to the apical surface (<30 µm; *Figure 5F*). In agreement with previous studies, we determined an average velocity of 86 ± 5 µm/s in standard bath solution (aCSF 3 $K^+$) (*Lechtreck et al., 2008*). ATP application (100 µM) slightly increased average bead velocity (96 ± 8.4 µm/s), but was not statistically significantly different. In increasing external [$K^+$], we observed a reduction in average velocity, which was reduced by preincubation with 10 µM nimodopine (79 ± 3 µm/s, aCSF 40 $K^+$; 67 ± 5.5 µm/s, aCSF 70 $K^+$; 80 ± 4.3 µm/s, aCSF 70 $K^+$ + nimodipine; *Figure 5G*). Note, however, that the velocity of individual beads varied considerably (*Figure 5G*). We cannot rule out an effect of $Ca_V$ channels in governing overall motile cilia function, but if it exists, is small and variable.

## Discussion

The 9 + 2 structural arrangement of microtubules and inherent dynein ATPase-activity which drives ciliary motion are conserved among motile cilia from protist to humans. However, $Ca^{2+}$ modification of cilia function in mammalian ependymal cells is quite distinct from that in protists (*Inaba, 2015*; *Satir and Christensen, 2007*). While beat reversal, and changes in waveform critically depend on the activity of ciliary $Ca_V$ channels in *Chlamydomonas* and *Paramecium* (*Fujiu et al., 2009*; *Kung and Naito, 1973*; *Matsuda et al., 1998*), we found little evidence linking $Ca^{2+}$ in ependymal cell bodies or motile cilia to their function. Moreover, unlike primary cilia in which specialized ciliary channels (polycystins) predominate (*DeCaen et al., 2013*; *Delling et al., 2013*), we found that $Ca_V$ channels in the cell body primarily determine changes in motile cilia [$Ca^{2+}$]. Although we could not accurately quantify the relative $Ca_V$ channel densities in motile cilia compared to the cell body or test the possibility of non-uniform channel distribution along the cilium shaft, these channels were not enriched at the ciliary tip. Whether sparsely ciliated cells are less well differentiated or represent a subclass of ciliated ependymal cells is unclear, but the low abundance of $Ca_V$ channels revealed by patch clamping of these cilia is consistent with a cytoplasmic control of ciliary [$Ca^{2+}$] in imaging experiments from multiciliated cells. We thus conclude that ependymal cilia have distinct electrical properties as compared to flagella (cilia) that are used for locomotion in mammalian sperm (*Kirichok et al., 2006*; *Miki and Clapham, 2013*; *Qi et al., 2007*; *Ren et al., 2001*) and protists (*Beck and Uhl, 1994*; *Dunlap, 1977*; *Fujiu et al., 2009*; *Kung and Naito, 1973*; *Matsuda et al., 1998*).

Based on direct measurements of cell body and motile cilia membrane potentials, and voltage-activated $Ca_V$ channels in both compartments, we hypothesize that the motile cilium compartment is

passively coupled and primarily controlled by channel activity and [Ca$^{2+}$] in the cell body, rather than being independently regulated. Thus, a cellular response would promote simultaneous signaling to all cilia, perhaps slowly entraining the cilia and enabling signal propagation to neighboring cells (*Sanderson et al., 1988*; *Schmid and Salathe, 2011*). Unlike primary cilia, ependymal cilia had only slightly elevated resting [Ca$^{2+}$], thus enabling smaller changes in cytoplasmic Ca$^{2+}$ to equilibrate within cilia. Indeed, membrane depolarization triggered activation of Ca$_V$ channels, which resulted in a robust change in ciliary [Ca$^{2+}$]. While we could not differentiate the onset of the Ca$^{2+}$ response in the cilium from that in the cell body, the results suggest that ciliary [Ca$^{2+}$] follows cytoplasmic [Ca$^{2+}$].

Previous studies suggest that different sources of Ca$^{2+}$ (internal stores, cell membrane flux) can modify the beat frequency of mammalian motile cilia (*Di Benedetto et al., 1991*; *Lansley et al., 1992*; *Nguyen et al., 2001*; *Salathe and Bookman, 1995*; *Schmid and Salathe, 2011*). ATP-activated P2X7 receptor-mediated Ca$^{2+}$ influx made a minor contribution to the ATP-induced increase in beating frequency of ependymal cilia, while activation of the adenosine A2B receptor made a more significant change, perhaps via slower G protein-dependent pathways (*Genzen et al., 2009b*). In contrast, we observed that membrane depolarization, or application of ATP, did not significantly alter ependymal motile cilia function. We observed only a slight decrease in bead velocity in the lateral ventricles of acute brain slices that correlated with depolarization and Ca$_V$ channel activation. Since we did not detect a change in ciliary beating frequency in in vitro cultures of ependymal cells, we suspect that coupling to other cell types or factors mediates the observed reduction. The variability in velocities of individual beads, however, leads us to question the physiological relevance of the observed changes. Finally, although previous studies suggest that ependymal cell differentiation in vitro reflects in vivo conditions (i.e. postnatal maturation into multiciliated cells [*El Zein et al., 2009*; *Guirao et al., 2010*; *Spassky et al., 2005*]), we cannot exclude the possibility of altered channel expression. Since we did not observe significant changes in ciliary beating in response to membrane depolarization in either acute brain slices or in vitro cultures, we suspect that control mechanisms are similar.

As molecular oars, 9 + 2 motile cilia power fluid flow across the surface of epithelia. Defects in cilia motility are linked to primary ciliary dyskinesia (PCD) and manifest in disease conditions such as chronic sinusitis, male infertility, and hydrocephalus (*Ibanez-Tallon et al., 2003*; *Roy, 2009*). Here we have examined the electrophysiological and Ca$^{2+}$ signaling properties of motile cilia in ependymal cells. We conclude that they are quite distinct from those of primary cilia (*DeCaen et al., 2013*; *Delling et al., 2013*) and the flagella that power motile cells. First, membrane potential and [Ca$^{2+}$] closely follow those in the cytoplasm. This stems naturally from a relative paucity of ion channels in the cilia compared to the nonselective channels in primary cilia (*DeCaen et al., 2013*). Second, voltage-dependent calcium channels change [Ca$^{2+}$] in the cytoplasm that propagates readily into the motile cilia, but only marginally alters beat frequency or fluid flow. These findings, although they cannot necessarily be generalized to all motile cilia, suggest that Ca$_V$ channels in ependymal cells primarily serve other secretory or differentiation roles in the cell body.

## Materials and methods

### Mice

Transgenic *Arl13B-EGFP$^{tg}$* (*Delling et al., 2013*) and C57BL/6 (wild type) mice were used in this study. Animal research protocols were approved by the IACUC of Boston Children's Hospital.

### Cell culture

#### Ependymal cell culture

Primary ependymal cell cultures were established as described previously (*Delgehyr et al., 2015*; *Guirao et al., 2010*). Briefly, brains of postnatal day 0–3 transgenic *Arl13B-EGFP$^{tg}$* mice (*Delling et al., 2013*) or wild type mice were collected, telencephala dissected, the choroid plexus and meninges removed, the tissue enzymatically digested, cells mechanically dissociated, plated and grown to confluence. Confluent cell layers were trypsinized, and replated on poly-L-lysine coated coverslips at a density of 10$^7$ cells/ml in Dulbecco's modified Eagle's medium (DMEM) with high glucose, GlutaMAX supplement, and pyruvate (Life Technologies), 10% decomplemented fetal bovine serum (FBS, Atlanta Biologicals), and 100 units/ml penicillin/100 μg/ml streptomycin (Atlanta

Biologicals), which was replaced by serum-free media the next day. Ependymal cells were cultured for up to two weeks.

## Cell lines

HEK293 and mIMCD3 cells (ATCC) were cultured in DMEM/F12 (Life Technologies) with 10% FBS and 100 units/ml penicillin/100 µg/ml streptomycin.

## Immunostaining

Paraformaldehyde (PFA, 4%) fixed frozen sections of brain ventricles (from a juvenile male) and cultured ependymal cells were permeabilized with 0.1–0.5% Triton X-100, blocked in 5% NGS, 1% BSA, 0.1% Triton X-100, 0.05% Tween20, and incubated with unconjugated Fab fragment goat anti-mouse (Jackson Immuno Research) to block endogenous mouse IgGs. Primary antibodies (goat anti-GFP FITC conjugate, Novus Biologicals or rabbit anti-GFP Alexa Fluor 488 conjugate, Life Technologies; rat anti-mCherry, Life Technologies; mouse anti-acetylated tubulin, Sigma-Aldrich; rabbit anti-Spag6, Sigma-Aldrich) were applied overnight at 4°C. Sections and cells were washed in PBS-T and incubated with secondary antibody (Alexa Fluor 568 or 647 conjugates, Life Technologies and Jackson Immuno Research). Nuclei were stained with Hoechst 33342 (Life Technologies) and sections/cells mounted in Prolong Diamond Antifade (Molecular Probes). Cells were cultured on glass coverslips (VWR) or gridded glass-bottom dishes (Ibidi) for staining. mIMCD3 cells were fixed and stained accordingly and images acquired with settings in the range of the acquisition settings used for Spag6-stained ependymal cells. All images were acquired with a FluoView1000 confocal microscope (Olympus) and processed using ImageJ (NIH).

## Electrophysiology

### Data acquisition

Recordings from cultured Arl13b-EGFP expressing primary ependymal cells (cell bodies and motile cilia) were typically performed at 7–14 days in vitro (DIV) after serum starvation at room temperature. Electrodes were pulled from borosilicate glass (Sutter Instruments) with an inner diameter of 0.75 mm or 0.86 mm and a resistance of ~16–30 MΩ or 4–8 MΩ for motile cilia and cell body recordings, respectively. All data were acquired using an Axopatch 200B amplifier (Molecular Devices). Whole-cell and single-channel currents were digitized at 20 kHz and filtered at 5 kHz or 2 kHz, respectively. For capacitive current analysis, currents were digitized at 50 kHz and filtered at 10 kHz. A voltage protocol was applied for motile cilium break-in. Liquid junction potentials were measured using a salt bridge filled with 3 M KCl solution, as described by Neher (*Neher, 1992*). Voltages were corrected for the liquid junction potential between the bath and pipette solution before acquisition. A 150 mM KCl agar bridge was used in all other recordings. Membrane capacitance and series resistance were not compensated in whole-motile-cilium/whole-cell recordings. Electrodes were routinely Sylgard-coated in single-channel recordings. The image series illustrating seal formation (see *Video 2*) and the image showing dye diffusion (50 µM Alexa Fluor 594 hydrazide, Life Technologies) from the recording pipette into a motile cilium were captured with an Evolution QEi monochrome camera (Media Cybernetics).

### Data analysis

Data were analyzed using Clampfit (Molecular Devices), Origin (OriginLab) and IgorPro (Wavemetrics). Membrane capacitance and series resistance were determined from capacitive currents obtained from hyperpolarizing voltage steps (-80 mV to -100 mV). Briefly, the steady state current was subtracted and the transient current integrated to determine the transferred charge, Q. The membrane capacitance ($C_m$) was then calculated as $C_m = Q/\triangle V$, where $\triangle V$ is the size of the voltage step. The series resistance ($R_{series}$) was determined from $C_m$ and time constant ($\tau$) as $R_{series} = \tau/C_m$.

Voltage-gated calcium channel recordings were analyzed after removal of passive components resulting from depolarizing voltage steps. Three consecutive current responses to a hyperpolarizing voltage step (20 mV or 10 mV in case of nimodipine) were recorded before each voltage pulse. The current responses to the hyperpolarizing pulses were summed, inverted and scaled, and then subtracted from the current response to a given depolarizing voltage step. High series resistance, as observed in whole-motile-cilia recordings, results in slow or incomplete voltage clamp of

membranes. Slower $V_m$ changes induced $Ca_V$ currents with a lag ($\tau = R_{series}C_m$, see exemplary trace *Figure 3—figure supplement 1B*), and $V_m$ becomes larger than the voltage command, introducing a voltage error that scales with the current amplitude ($V_m = V_p - I_mR_{series}$).

The displayed single channel recordings were filtered offline at 1 kHz and capacitive and leak currents subtracted. Histograms were plotted with either all points included or confined to open points (events with fractional amplitude of 0.9).

The total number of channels per cell was calculated as $N = I_m/P_o(i)$, where $I_m$ is the average peak current (recorded from ependymal cells in the presence of BayK8644 at 0 mV; 297.8 pA), $P_o$ is the single channel open probability (calculated from a single channel recording in a cell membrane patch by integrating 400 ms voltage pulses over 9 sweeps; 0.32), and (i) is the single channel amplitude at 0 mV (analyzed from the record shown in *Figure 3C*; 1.4 pA). The surface area of a motile cilium was estimated using the formula for a cylinder: $A = 2\pi r (r + h)$, where r is the radius and h is the height. If the average length of an ependymal motile cilium is ~11.5 µm (*Spassky, 2013*) and the diameter ~0.25 µm, then the surface area of an ependymal motile cilium is ~9.1 µm$^2$. Since $C_m$ is approximately 1 µF/cm$^2$ in biological membranes, the motile cilium capacitance is ~0.091 pF. The average cell capacitance was 18.5 pF (after gap junction uncoupling in TEA-Cl/BaCl$_2$ bath, see *Figure 2E*). The number of $Ca_V$ channels per motile cilium was estimated by multiplying the total number of channels per cell with the quotient of cilium to cell surface area.

The resistance of a motile cilium as an insulated cable was estimated using the equation $R = \rho l/A$, where $\rho$ is the resistivity (150 Ω·cm), l is the length of the cilium (11.5 µm) and A is the cross-sectional area ($\pi r^2$, with r = 0.125 µm). The actual internal resistivity of a motile cilium is unknown. The value for $\rho$ is based on the assumption that the internal resistivity of a motile cilium is similar to cytoplasmic resistivity.

## Solutions

The standard bath solution (aCSF) was (in mM): 148 NaCl, 3 KCl, 1.4 CaCl$_2$, 0.9 MgCl$_2$, 10 HEPES, 1 NaH$_2$PO$_4$; pH 7.3 with NaOH. The standard pipette solution was (in mM): 121.5 K-methanesulfonate, 9 KCl, 1.8 MgCl$_2$, 9 EGTA, 14 creatine phosphate (Na$^+$ salt), 4 MgATP, 0.3 NaGTP, 10 HEPES; pH 7.3 with KOH. For gramicidin-perforated patch recordings (current clamp), pipettes were filled with a solution containing (in mM): 145 KCl, 20 HEPES, 1 EGTA; pH 7.3 with KOH. A gramicidin stock solution was prepared just before use and added to the pipette solution at a final concentration of 50 µg/ml. The membrane potential was assessed in varying bath KCl by substituting NaCl with 20, 40, 70, and 140 mM KCl (denoted as aCSF 20 K$^+$, aCSF 40 K$^+$, aCSF 70 K$^+$, and aCSF 140 K$^+$). Cells were typically uncoupled by addition of flufenamic acid (FFA, 100 µM; Sigma-Aldrich) in whole-cell/whole-cilia recordings. FFA did not significantly block $Ca_V$ currents (*Guinamard et al., 2013*), since we recorded similar currents in trypsin-uncoupled cells (data not shown). K$^+$ channel conductances were reduced by perfusion of a tetraethylammonium based solution (in mM): 110 TEA-Cl, 30 BaCl$_2$, 10 HEPES; pH 7.3. The same bath solution was used to record voltage-gated calcium channels in the whole-cell/whole-cilium configuration. Cell-attached single channel currents were recorded in a KCl bath solution to null the membrane potential (in mM): 145 KCl, 20 HEPES, 1 EGTA; pH 7.3 with KOH. The pipette solution was (in mM): 110 BaCl$_2$, 10 HEPES; pH 7.3.

## Adenoviral vector

Expression clones were created by cloning mSSTR3-mCherry-GCaMP6s into pENTR 3C and subsequent LR recombination with the destination vector pAD/CMV/V5-DEST vector (Life Technologies). Glycine-serine linkers were introduced between mSSTR3-mCherry and mCherry-GCaMP6s as described previously (*Delling et al., 2013*). The expression vector was digested with *PacI* and transfected into HEK293 AD cells. The crude adenoviral stock was further amplified by infection of HEK293 cells. Cultured ependymal cells (derived from wild type pups) were typically transduced with the recombinant adenoviral vector (pAD-mSSTR3-mCherry-GCaMP6s) at DIV6 or 8 and functional expression of the sensor in motile cilia assessed at DIV10 to 12.

## Ca²⁺ imaging

### Calibration of the sensor

The calibration curve for the cilium-targeted ratiometric sensor (mSSTR3-mCherry-GCaMP6s) was obtained by incubating adenovirus-transduced ependymal cells in solutions with free $[Ca^{2+}]$ ranging from 50 nM to 10 µM. Digitonin was added at a final concentration of 16 µM to permeabilize the cells and images were acquired with a FluoView1000 confocal microscope (Olympus) for multiple cells after a 5 min permeabilization. The standard solution contained (in mM): 145 NaCl, 5 KCl, 5 EGTA, 10 HEPES, and $CaCl_2$; pH 7.3 with NaOH. MaxChelator (maxchelator.stanford.edu) was used to calculate the total $CaCl_2$ needed to obtain the desired free $[Ca^{2+}]$. The acquisition parameters were initially established for GCaMP6s and mCherry fluorescence in the solution containing the highest free $[Ca^{2+}]$ and then held constant across experiments. Non-sequential scan mode was chosen to avoid offsets in the position of beating cilia (basal conditions). Bleed-through of GCaMP6 fluorescence emission into the Cherry signal was subtracted as explained below. Image analysis was performed using ImageJ (NIH) and data plotted using Origin (OriginLab). Images were background-subtracted and a mask created by thresholding the Cherry channel to exclude pixels with no sensor fluorescence. The Cherry mask was then applied to the images with GCaMP6s fluorescence and the resulting image divided by the background-subtracted image of the Cherry channel to obtain ratioed images. Saturated points were subtracted by thresholding the Cherry channel before background subtraction and multiplying it by the ratio image. The mean ratio of F_GCaMP6s and F_mCherry for various free $[Ca^{2+}]$ was measured by averaging the fluorescence intensity of all motile cilia per cell, of multiple cells. To correct for bleed-through, images of HEK293 cells expressing GCaMP6s (in pcDNA3.1) were acquired upon stimulation with 2 µM ionomycin in varying external $CaCl_2$ concentrations, using the identical acquisition settings as for the acquisition of the calibration curve. Images were background-subtracted and ROIs selected in the cytoplasmic compartment. The GCaMP6s fluorescence was plotted as a function of mCherry fluorescence and the data points fitted linearly. The resulting slope accounts for the bleed-through and was subtracted using the following formula: $A_c = F\_GCaMP6s/F\_mCherry - \alpha GCaMP6s$, where $A_c$ is the corrected average ratio and $\alpha GCaMP6s$ is the slope of the fit. The corrected ratios were then plotted as a function of free $[Ca^{2+}]$ and the data points fitted with the Hill equation. The basal free $[Ca^{2+}]$ in motile cilia was calculated from the average of the corrected fluorescence ratio in standard bath solution (aCSF) and the parameters from the Hill fit of the calibration curve.

### Ca²⁺ imaging of motile cilia

Motile cilia expressing mSSTR3-mCherry-GCaMP6s were imaged non-sequentially using a FluoView1000 confocal microscope (Olympus). Images were acquired with frame rates of 0.58 s (ionomycin) and 1.109 s (depolarization). Cells were imaged in aCSF standard bath solution (see Electrophysiology, *Solutions*). NaCl was substituted with KCl in solutions with 40, 70, and 140 mM KCl (denoted as aCSF 40 $K^+$, aCSF 70 $K^+$, and aCSF 140 $K^+$). The $Ca^{2+}$-free external solution contained (in mM): 148 NaCl, 3 KCl, 2.3 $MgCl_2$, 10 HEPES, 1 $NaH_2PO_4$, and 0.5 EGTA; pH 7.3 with NaOH. Solution exchange was achieved by gravity driven bath perfusion (5–10 s delay). F_GCaMP6s/F_mCherry ratio changes were quantified from the mean fluorescence intensity of all motile cilia per cell (ImageJ), normalized to the initial ratio, and plotted as a function of time (Origin). Ratios were not corrected for bleed-through (see above) as settings were slightly adjusted for each channel prior to imaging experiments. Displayed ratios are thus lower than the actual ratios (linear correlation between F_GCaMP6s and F_bleed-through, see *Figure 4—figure supplement 1*). In experiments in which changes in cytoplasmic $[Ca^{2+}]$ was assessed, ependymal cells were loaded with Oregon Green 488 BAPTA-1 AM (OGB-1, 2 µm) for 1 hr at room temperature in aCSF. Changes in OGB-1 fluorescence intensity were measured using ImageJ.

### Ca²⁺ diffusion into motile cilia

For $Ca^{2+}$ diffusion experiments, mSSTR3-mCherry-GCaMP6s-transduced ependymal cells were loaded with O-nitrophenyl EGTA-AM (NP-EGTA, 6 µM) for 1 h at room temperature in aCSF. A brief 405 nm laser pulse (200 ms) was applied to uncage $Ca^{2+}$ in the cytoplasm. Subsequent changes in ciliary $[Ca^{2+}]$ were imaged at a frame rate of 0.129 s. To simultaneously monitor changes in cytoplasmic and ciliary $[Ca^{2+}]$, ependymal cells were loaded with NP-EGTA (6 µm) and OGB-1 (2 µm) and

imaged from the side as follows: Cells were grown on polyester membrane inserts (Transwell, Costar), the membrane cut, flipped, pinned down, and covered by a self-adhering cover with dual ports for solution exchange (180 µl volume, Electron Microscopy Sciences). Ciliary beating was reduced by continuous perfusion of 100 µM sodium metavanadate (SMVD) for 5 min. Cells with halted ciliary beating were chosen for the experiments. The uncaging stimulus (405 nm, 200 ms laser pulse) was focused to the ciliary base and diffusion measured by scanning a single line from the cytoplasm to the tip of a motile cilium. A few recordings displayed two apparent velocities, presumably related to the different kinetics of OGB-1 and GCaMP6s. In those cases, we analyzed the velocity for the distal part of the cilium. Non-sequential image series were also acquired at a frame rate of 0.254 s.

## High speed imaging of motile cilia

Cilia motility and fluid flow were assessed at 60x magnification on an inverted Olympus IX70 microscope. Images were acquired with an OrcaFlash 4.0 high speed camera and HCImageLive imaging software (Hamamatsu).

### Analysis of beating frequency

Cultured ependymal cells (DIV12) were imaged at room temperature at 200 frames/s for 2 s in aCSF (see Electrophysiology) pre- and post- incubation with SMVD (100 µM, 5 min). Acquisition areas (typically several per condition and coverslip) were chosen randomly. The beating frequency was analyzed from kymographs (*Lechtreck et al., 2008*) created in ImageJ using the re-slice function and counting peaks/s. Typically 1–3 cilia were analyzed per acquisition area. For comparison of cilia beat frequencies from cultured multi- and sparsely-ciliated ependymal cells (DIV10), images were acquired at 100 frames/s for 4 s. In experiments in which cilia motility was assessed in response to aCSF 70 $K^+$ or aCSF 140 $K^+$, images were captured at a frequency of 100 Hz for 30 s. Gravity-driven bath perfusion was used to continuously perfuse cells with aCSF (controls) or to exchange the bath solution to aCSF 70 $K^+$ or aCSF 140 $K^+$. Solution exchange was initiated immediately before image acquisition to capture the transition (5–10 s perfusion delay). Kymographs were plotted for 2 cilia per recording and beating frequencies analyzed at 3 different time points (0–1 s, 15–16 s and 29–30 s).

### Analysis of bead velocity

Brain hemispheres of postnatal day 10–14 wild type or transgenic *Arl13B-EGFP*[tg] mice (*Delling et al., 2013*) were sectioned with a vibratome (Leica) to 200 µm slices in an ice-cold oxygenated solution containing in mM: 87 NaCl, 25 $NaHCO_3$, 1.25 $NaH_2PO_4$, 2.5 KCl, 75 sucrose, 25 glucose, 7.5 $MgCl_2$. Slices were stored on ice in an oxygenated solution containing in mM: 87 NaCl, 25 $NaHCO_3$, 1.25 $NaH_2PO_4$, 2.5 KCl, 75 sucrose, 25 glucose, 7 $MgCl_2$, 0.5 $CaCl_2$ until use (typically 1–3 h after slicing). Recordings were performed at room temperature in an enclosed 180 µl perfusion chamber (Electron Microscopy Sciences). All aCSF-based solutions (see Electrophysiology) used for brain slice imaging additionally contained 10 mM glucose. Slices were routinely perfused for 2 min with aCSF prior to the perfusion of polystyrene beads (500 nm in diameter, Sicastar-greenF, Micromod) in the respective solution for 30 s. The perfusion was subsequently stopped to record motile cilia driven bead movements along the surface of the lateral ventricle (4 s at 100 Hz). Some slices were perfused with beads in aCSF control solution and, *e.g.*, aCSF 70 $K^+$, and slices washed for 2 min with aCSF between recordings. In experiments with nimodipine, slices were perfused for 2 min with aCSF plus nimodipine (10 µM) prior to bead perfusion in aCSF 70 $K^+$ plus nimodipine. Bead velocities were analyzed using the ImageJ manual tracking plugin. Only beads that could be tracked for at least 4 frames were included in the analysis.

## RT-PCR

RNA was isolated from ependymal cell cultures and dissected tissue (brain, heart, eyes and skeletal muscle) using TRIzol reagent (Life Technologies). Ependymal cell cultures were derived from a transgenic *Arl13B-EGFP*[tg] mouse pup and tissue collected from a 2 week-old heterozygous *Arl13B-EGFP*[tg] male. cDNA was synthesized from total RNA using Superscript III (Life Technologies). The reverse transcriptase was omitted from the cDNA synthesis reaction in controls for cultured

ependymal cells. PCR products were amplified with Phusion high-fidelity DNA polymerase (NEB) using a touchdown PCR protocol with decreasing annealing temperatures (3 cycles at 68, 65, and 62 degrees; and 25 cycles at 58 degrees). The following intron-spanning primers were used:

$Ca_V1.1$ (Cacna1s), fw 5′ ATGACAACAACACTCTGAACCTC 3′ and rv 5′ GGAAGCCGTAGGCTATGATCT 3′ (PrimerBank ID 189409135c2); $Ca_V1.2$ (Cacna1c), fw 5′ CCTGCTGGTGGTTAGCGTG 3′ and rv 5′ TCTGCCTCCGTCTGTTTAGAA 3′ (PrimerBank ID 192322a1); $Ca_V1.3$ (Cacna1d), fw 5′ GCTTACGTTAGGAATGGATGGAA 3′ and rv 5′ GAAGTGGTCTTAACACTCGGAAG 3′ (PrimerBank ID 134288874c2); $Ca_V1.4$ (Cacna1f), fw 5′ TACTAATCCCATTCGTCGGTCC 3′ and rv 5′ CATAGGCTACGATCTTGAGCAC 3′ (PrimerBank ID 115648150c2); GAPDH, fw 5′ TGGCCTTCCGTGTTCCTAC 3′ and rv 5′ GAGTTGCTGTTGAAGTCGCA 3′ (PrimerBank ID 126012538c3).

A 2-Log DNA ladder (NEB) and PCR samples were loaded on 1% NuSieve 3:1 agarose gels (Lonza) and images acquired with the Syngene GBox system and processed using ImageJ. The PCR reactions for $Ca_V1.1$ and its controls were run separately and a larger volume (1.5x) was loaded on the gel (weak amplification of $Ca_V1.1$ transcripts from control tissue).

## Acknowledgements
We thank N Blair for insightful discussions and assistance with data analysis. We also thank S Febvay for advice with image analysis and members of the Clapham lab for discussion.

## Additional information

### Competing interests
DEC: Reviewing editor, *eLife*. The other authors declare that no competing interests exist.

### Funding

| Funder | Author |
|---|---|
| Howard Hughes Medical Institute | David E Clapham |

The funders had no role in study design, data collection and interpretation, or the decision to submit the work for publication.

### Author contributions
JFD, Conception and design, Acquisition of data, Analysis and interpretation of data, Drafting or revising the article; MD, Conception and design, Analysis and interpretation of data, Drafting or revising the article, Contributed unpublished essential data or reagents; DEC, Conception and design, Analysis and interpretation of data, Drafting or revising the article

### Ethics
Animal experimentation: Animal Resources at Children's Hospital (ARCH) is AAALAC accredited. All animals were handled according to the protocol (#13-03-2380R) approved by the Institutional Animal Care and Use Committee (IACUC) of Boston Children's Hospital.

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
