## [Decision Letter]

Thank you for submitting your work entitled "Ion channels and calcium signaling in motile cilia" for consideration by *eLife*. Your article has been reviewed by three peer reviewers, and the evaluation has been overseen by Richard Aldrich as the Senior Editor and Reviewing Editor.

The reviewers have discussed the reviews with one another and the Reviewing editor has drafted this decision to help you prepare a revised submission.

As the three reviewers find the work to be convincing and important, and they only request minor revisions, the separate verbatim reviews are enclosed.

Please respond to their comments when submitting your revision.

Reviewer #1

Doerner et al. report the first comprehensive patch clamp analysis of "typical" mammalian motile cilia. In contrast to those of whole cell plasma membranes, which have been studied for >70 years, the biophysical properties of organelle membranes are generally very little understood, partially because of the difficulty of applying electrophysiological recording techniques. The authors overcame the difficulty by developing a quite neat set of tools, and painstakingly recorded whole-cilium currents from ependymal cells. Two of the major findings in the paper are quite surprising. First, the biophysical properties of the motile cilia are intriguingly very different from those of primary cilia that the authors characterized before. For example, the motile cilia do not seem to have the PKDL Ca^2+^-permeable channels abundantly present in primary cilia. In fact, the motile cilia don't seem to have significant number of channels at all. The apparent lack of major ion channels in the cilia is consistent with the authors' finding of tight electrical and chemical (Ca^2+^-diffusion) coupling between the cilia and cell body. Second, the beating of the cilia do not seem to be affected by [Ca^2+^] increases. This again is quite surprising given that most of us would expect the beating of 9+2 -structured cilia to be like that of e.g. the flagella of sperm cells found in both vertebrates and invertebrates and to be highly influenced by Ca^2+^.

The experiments and the quantitative analysis in the paper are well done and the conclusions are reasonably convincing. The results are novel and surprising. I do not have any major criticism.

Minor suggestions:

1) Because it's not possible to record motile cilia pulled off from cell bodies, like the authors have previous done for sperm flagella and primary cilia, the authors used cilium-attach to record cilium-restricted channels. The recordings (Figure 3) used BaCl_2_-containing pipette that is presumably optimized for Ca_v_ channels. Have the authors tried similar cilium-attach recordings with other pipette solutions more suitable for other channels?

2) Figure 1, it's surprising that the only currents (presumably from whole cell and whole cilium) are the "leak-like" K^+^ currents. Did the authors test voltages beyond +60 mV? The outward PKD-like currents reported in the DeCaen 2013 paper appear to be more obvious in voltages of ~ +50 to +100 mV.

3) Figure 1, the authors might want to indicate E_K_ values on the X-axis for each [K^+^] condition.

4) Figure 3, the PCR reactions for Ca_V_1.2 & Ca_V_1.3 need negative controls if available.

5) Paragraph four, subheading “Data Analysis”, "where ρ is the resistivity (150Ω.cm)". It's not clear to me where the cilium resistivity number is from.

Reviewer #2

This important manuscript uses ciliary patch-clamp electrophysiology and optical methods to provide one-of-a-kind insight into the mechanism of Ca^2+^ signaling within the motile mammalian cilia. Because the same laboratory has previously provided similar characterization for flagellum and primary cilium, they are uniquely positioned to provide a comparison between these three ciliary structures. This reviewer thinks that the data and conclusion contained within the manuscript are solid, but would like to obtain answers/comments to the following points:

1) Subheading “Ependymal motile cilia identification and patch clamp”: It would be appropriate to provide an explanation of why, from the authors' point of view, seals were formed much easier with sparsely ciliated cells. Do you think that slower motility of the cilia of sparsely ciliated cells makes them not representative of normal motile cilia? How do beat frequencies of normal and sparsely ciliated cell compare?

2) Paragraph three, subheading “Ependymal motile cilia identification and patch clamp”: Would not it be better to say "carried by K^+^" rather than "K^+^-dependent" to indicate the fact that the reversal potentials for the current seem to be very close to K^+^ Nernst potentials?

3) Figure 2: Why does the capacitance transient recorded through the cilium does not saturate at the zero current level?

4) Why is the calculated ciliary resistance (~350 MΩ, paragraph two, subheading “Electrical coupling of motile cilia with the cellular compartment”) twice as low as measured (~180 MΩ,)? Which one should the reader take as the actual ciliary resistance?

5) Paragraph two, subheading “Ca_v_-mediated currents in ependymal cells”: Were all the single-channel recordings from cilia made from the very tip (“Typically, we patched the bulging tip of a motile cilium)? If yes, is there a possibility that Ca_v_ channels (or other Ca^2+^ channels) are abundant on the sides of the cilium?

Reviewer #3

Doerner et al. present a thorough analysis of calcium physiology in ependymal cell cilia and examine its potential role in regulating cilia beat frequency. Previous studies linking calcium signaling to control of ciliary beat frequency examined airway and reproductive tract cilia and the role of TRP channel-mediated calcium influx. Prior to this paper, not much was known about ependymal cilia and calcium.

The authors successfully patch clamp motile cilia and demonstrate convincingly that they are well coupled electrically to the cytoplasm. They determine the resting calcium level and show it is different from primary cilia (lower), further refining the distinct features of these different types of cilia. They also show that elevation of cytoplasmic calcium is rapidly transmitted to the motile cilia axoneme implying an absence of any barrier to diffusion at the cilia base. Voltage dependent changes in motile cilia calcium were mediated by L-type calcium channels, distinct from TRP channel currents seen in primary cilia. Finally the authors indirectly confirm their findings in brain slices using bead movement as a surrogate for cilia beat frequency.

Overall the studies are rigorously performed and, although results are negative with regard to calcium control of cilia beat in ependymal cells, add significantly to our understanding of motile cilia heterogeneity and the differences between primary and motile cilia.

Concerns to address in the Discussion:

The studies were necessarily done in cell culture. It is possible that the state of cell differentiation of ependymal cells could be altered by time spent in vitro. In vivo control mechanisms (channel expression) could be altered. The authors should state any evidence they have that this is not the case, i.e. that the in vitro ependymal cell phenotype reliably reflects what is seen in vivo.

Sparsely ciliated cells were primarily used for patch clamping studies. Again, an implication here is that these cells might be less well differentiated than densely ciliated cells and may possibly lack in vivo control mechanisms; Some statement acknowledging (or refuting) this possibility should be added to the Discussion.

---

## [Author Response]

Reviewer #1 *1) Because it's not possible to record motile cilia pulled off from cell bodies, like the authors have previous done for sperm flagella and primary cilia, the authors used cilium-attach to record cilium-restricted channels. The recordings (Figure 3) used BaCl_2_-containing pipette that is presumably optimized for Ca_v_ channels. Have the authors tried similar cilium-attach recordings with other pipette solutions more suitable for other channels?*

We mainly focused on Ca_V_ single channel currents using BaCl_2_ in the pipette solution and voltage protocols optimized for Ca_V_ channel recordings. Many TRP channels, including PKD2-L1 channels conduct Ba^2+^ (DeCaen et al., 2013). If PKD2-L1 channels were enriched in motile cilia as they are in primary cilia, we should have observed single channel openings at hyperpolarized potentials (we did not). If the reviewer is referring to the possibility that K^+^ channels may be blocked in presence of Ba^2+^, we used K^+^- and Na^+^-based pipette solutions in some experiments and occasionally observed single channel openings, indicating the presence of other ion channels in motile cilia. While we have not extensively tested the presence of ligand-gated ion channels such as P2X7 receptors, the small number of patches in which we observed single channel openings (with different pipette solutions) suggests that motile cilia have many fewer ion channels than primary cilia. *2) Figure 1, it's surprising that the only currents (presumably from whole cell and whole cilium) are the "leak-like" K^+^ currents. Did the authors test voltages beyond +60 mV? The outward PKD-like currents reported in the DeCaen 2013 paper appear to be more obvious in voltages of ~ +50 to +100 mV.*

We did not extensively test voltages beyond +60 mV. The observed leak-like K^+^-conductance obscures identification of currents. For example, the outwardly rectifying TEA-sensitive K^+^ current in ependymal cells became evident only after cell uncoupling by flufenamic acid. While flufenamic acid and other fenamates reversibly block gap junctions (ependymal cells grow in a monolayer), they also modulate (i.e. activate/inhibit) many ion channels (Harks et al., 2001; Guinamard et al., 2013). The electrical coupling between motile cilia and cellular compartments precludes more quantitative comparison. This would have been possible if motile cilia could be detached from the cell without rupture, but this proved impossible despite many attempts. *3) Figure 1, the authors might want to indicate E_K_ values on the X-axis for each [K^+^] condition.*

Thanks for the suggestion. We now indicate E_K_ values in Figure 1. *4) Figure 3, the PCR reactions for Ca_V_1.2 & Ca_V_1.3 need negative controls if available.*

We repeated the PCR reactions to include negative controls for ependymal cell cDNA (synthesis reaction minus reverse transcriptase) and replaced the image in Figure 3. *5) Paragraph four, subheading “Data Analysis”, "where ρ is the resistivity (150 ohm.cm)". It's not clear to me where the cilium resistivity number is from.*

The intrinsic internal resistivity of cilia is unknown. The value for ρ (150 Ω·cm) is based on the assumption that the internal cilia solution resistivity is similar to cytoplasmic resistivity (basically, physiological saline). We state this in the Methods. Reviewer #2*1) Subheading “Ependymal motile cilia identification and patch clamp”: It would be appropriate to provide an explanation of why, from the authors' point of view, seals were formed much easier with sparsely ciliated cells. Do you think that slower motility of the cilia of sparsely ciliated cells makes them not representative of normal motile cilia? How do beat frequencies of normal and sparsely ciliated cell compare?*

We now explain why sparsely-ciliated-cell motile cilia were most amenable for patch clamp. The main obstacles for patch clamping motile cilia are visualization, motility and multiciliation. Approaching a heavily ciliated cell with the patch pipette and applying light suction typically attracted more than one cilium to the pipette tip preventing seal formation on a single cilium. While we cannot exclude the possibility that cilia of sparsely ciliated cells present a subclass of cilia, we show that they retain motility and stain positive for a central pair marker, Spag6. We also carried out new experiments in which we compared the beat frequencies of cilia of in vitro cultured sparsely- and multiciliated ependymal cells. The cilia beat frequency of sparsely ciliated cells is ~2-3 fold less than that of multiciliated cells – we now state this in the text. Many groups have shown that extracellular hydrodynamic forces synchronize the movement of individual motile cilia in the group into metachronal waves. Cilia of sparsely ciliated cells experience different fluid dynamics (reduced flow) as compared to multiciliated cells, which likely affects their beat coordination and motility (Guirao & Joanny 2007, Guirao et al., 2010).

*2) Paragraph three, subheading “Ependymal motile cilia identification and patch clamp”: Would not it be better to say "carried by K^+^" rather than "K^+^-dependent" to indicate the fact that the reversal potentials for the current seem to be very close to K^+^ Nernst potentials?*

Certainly correct. We changed the wording accordingly.*3) Figure 2: Why does the capacitance transient recorded through the cilium does not saturate at the zero current level?*

We assume the reviewer means that the apparent capacitance current does not return to zero. This is because the time-independent current is seal leak in this trace. Despite bath substitution with TEA-Cl/BaCl_2_, the resistance was often < 1GΩ in whole-motile cilium recordings (seal leak after break-in). We now refer to the leak in the figure legend.*4) Why is the calculated ciliary resistance (~350 M*Ω*, paragraph two, subheading “Electrical coupling of motile cilia with the cellular compartment”) twice as low as measured (~180 M*Ω*,)? Which one should the reader take as the actual ciliary resistance?*

Neither represents the actual cilia resistance – they are boundaries and neither is terribly correct. The calculated (theoretical) ciliary resistance is an estimate based on the average dimensions of an ependymal motile cilium and the assumption that the cilium is a perfectly insulated cable with an internal resistivity of 150 Ω·cm. The measured resistance is the series resistance. Series resistance is the sum of the pipette resistance in series with the resistance through the cable-like cilium into the cell (essentially an extension of the pipette). *5) Paragraph two, subheading “Ca_v_-mediated currents in ependymal cells”: Were all the single-channel recordings from cilia made from the very tip (“Typically, we patched the bulging tip of a motile cilium”)? If yes, is there a possibility that Ca_v_ channels (or other Ca^2+^ channels) are abundant on the sides of the cilium?*

We performed all single channel recordings from the tip of the cilium, and were unable to make seals on the sides of cilia due to their small diameter and geometry. We cannot exclude Ca_V_ channel expression along the sides or at the base of a motile cilium. We discuss this in the Discussion.Reviewer #3*The studies were necessarily done in cell culture. It is possible that the state of cell differentiation of ependymal cells could be altered by time spent in vitro. In vivo control mechanisms (channel expression) could be altered. The authors should state any evidence they have that this is not the case, i.e. that the in vitro ependymal cell phenotype reliably reflects what is seen in vivo.*

You are correct, although acute slices may also have altered conditions. The ideal would be to patch clamp ependymal cells in situ, but this is not possible. We discuss the possibility of differences in channel expression in the Discussion. *Sparsely ciliated cells were primarily used for patch clamping studies. Again, an implication here is that these cells might be less well differentiated than densely ciliated cells and may possibly lack in vivo control mechanisms; Some statement acknowledging (or refuting) this possibility should be added to the Discussion.*

We discuss this point in the Discussion and Results along with the point discussed with Reviewer #2 about synchronization of cilia that should modify their beat frequencies.